# Polyacrylic Acid Nanoplatforms: Antimicrobial, Tissue Engineering, and Cancer Theranostic Applications

**DOI:** 10.3390/polym14061259

**Published:** 2022-03-21

**Authors:** Hassan Arkaban, Mahmood Barani, Majid Reza Akbarizadeh, Narendra Pal Singh Chauhan, Sapana Jadoun, Maryam Dehghani Soltani, Payam Zarrintaj

**Affiliations:** 1Department of Chemistry, University of Isfahan, Isfahan 8174673441, Iran; hassan.arkaban@gmail.com; 2Medical Mycology and Bacteriology Research Center, Kerman University of Medical Sciences, Kerman 7616913555, Iran; 3Department of Pediatric, Amir Al Momenin Hospital, Zabol University of Medical Sciences, Zabol 9861663335, Iran; 4Department of Chemistry, Faculty of Science, Bhupal Nobles’s University, Udaipur 313002, Rajasthan, India; narendrapalsingh14@gmail.com; 5Department of Analytical and Inorganic Chemistry, Faculty of Sciences, University of Concepcion, Edmundo Larenas 129, Concepcion 4070371, Chile; sjadoun022@gmail.com; 6Department of Chemistry, Faculty of Science, Yazd University, Yazd 8915818411, Iran; maryam.dehghanisoltani@gmail.com; 7School of Chemical Engineering, Oklahoma State University, 420 Engineering North, Stillwater, OK 74078, USA; payam.zarrintaj@gmail.com

**Keywords:** polyacrylic acid (PAA), synthesize, polymerizations, antimicrobial, anticancer, biosensing

## Abstract

Polyacrylic acid (PAA) is a non-toxic, biocompatible, and biodegradable polymer that gained lots of interest in recent years. PAA nano-derivatives can be obtained by chemical modification of carboxyl groups with superior chemical properties in comparison to unmodified PAA. For example, nano-particles produced from PAA derivatives can be used to deliver drugs due to their stability and biocompatibility. PAA and its nanoconjugates could also be regarded as stimuli-responsive platforms that make them ideal for drug delivery and antimicrobial applications. These properties make PAA a good candidate for conventional and novel drug carrier systems. Here, we started with synthesis approaches, structure characteristics, and other architectures of PAA nanoplatforms. Then, different conjugations of PAA/nanostructures and their potential in various fields of nanomedicine such as antimicrobial, anticancer, imaging, biosensor, and tissue engineering were discussed. Finally, biocompatibility and challenges of PAA nanoplatforms were highlighted. This review will provide fundamental knowledge and current information connected to the PAA nanoplatforms and their applications in biological fields for a broad audience of researchers, engineers, and newcomers. In this light, PAA nanoplatforms could have great potential for the research and development of new nano vaccines and nano drugs in the future.

## 1. Introduction

A polymer consists of macromolecules; many tiny molecules are joined via covalent connections. Polymers are the most ubiquitous biomaterials, with uses ranging from contact lenses to pharmaceutical carriers to implantation, artificial organs, tissue engineering, medical instruments, and Cancer theranostic [1]. Polymers are divided into synthetic and natural origins, and both branches have broad applications. This is owing to polymers’ unusual features, which established an altogether new notion when they were first presented as biomaterials [2,3,4]. Polymerizations are classified according to the reactions that occur throughout the synthesis process. It can be divided into three categories: addition, condensation, and metathesis polymerization [5,6]. Over the first time, a polymer used for construction purposes was engineered to be entirely resorbed and weaken over time. This approach was effectively implemented for the first time with catgut sutures, then afterward on bone fixation, ligament augmentation, plates, and pins, with questionable outcomes. Synthetic polymers have several characteristics, including low density, the potential to tailor aspects to various uses, water, chemical stability, and easy practicability [7].

Polyacrylic acid (PAA), formerly recognized as poly 1-carboxyethylene, is a high molecular weight synthetic (manufactured) polymer made with acrylic acid monomers. Poly (1-carboxyethylene) is a commercialized polymer at a small price. It is a biocompatible superabsorbent polymer soluble in water, nonpoisonous, and recyclable [8,9]. Superabsorbent polymers (SAP) are hydrophilic net-organized polymers like carboxylic acid, hydroxyl, and amines. Compared to typical water-absorbing polymers, superabsorbents can absorb a high volume of water and eliminate it even under pressure. Superabsorbents are extensively applied in healthcare, farming, agriculture, biomedical and everyday physiological goods, isolation techniques, and sewage treatment because of their exceptional qualities [10,11].

A stable structure can be formed by cross-linking poly (acrylic acid) [12,13]. Poly(acrylic acid), a common pH-responsive polymer [14], has typically served as a hydrophilic section for amphiphilic or amphipathic block copolymers with a variety of unique characteristics [15]. This polymer is currently sold as a soft white powder. It has the potential to generate translucent, fragile films. It is a hygroscopic polymer that can absorb and keep water molecules by absorption or adsorption from the environment. The glass transition temperature of pristine PAA is more than 100 °C. Polyacrylic acid acts as an anionic polymer in water [16]. Radiation, allyl ethers of hydrocarbons, and other chemical substances can crosslink poly (acrylic acid). At temperatures above 200 °C, PAA can also be cross-linked. Polyacrylic acid may release water and form an insoluble cross-linked network at high temperatures. The cross-linked PAA may create a gel-like structure. The case of poly (acrylic acid)-graft chitosan is an example of chemical cross-linking [17]. Block copolymers can be made by copolymerizing poly (acrylic acid) with other polymers. Various polymers, including polyacrylamide, polyethylene oxide, cellulose, and others, can form hydrogen-bonded complexes with PAA.

This artificial polymer is used for dispersion for many other purposes [18,19]. It is also utilized as a food supplement owing to low cytotoxicity. Because it is fully biodegradable, poly (acrylic acid) is of particular interest [20]. It has high adhesive strength due to its carboxylic acid activity [21]. It is also being employed in drug delivery systems [22]. It is also an eco-friendly polymer with superior mechanical strength and clarity [23,24]. As a result, PAA is employed in adhesives, coatings, homes, packaging, pharmacology, and other medical and biological industries [25]. Because of its outstanding properties and high water absorption, poly (acrylic acid), in either linear or cross-linked form, is widely used in several kinds of fake tears [26,27]. New progresses in science, bioinformatics and nanomedicine have important influence on human healthiness [28,29,30,31,32,33]. Nanotechnology is the manipulation of matter on a near-atomic scale to produce new structures, materials and devices [34,35,36]. The technology promises scientific advancement in many sectors such as medicine, consumer products, energy, materials and manufacturing [37,38,39]. The combination of nanotechnology and PPA advantages can help to develop more efficient nanocarriers. This review looked at various fascinating applications for polyacrylic acid nanoplatforms, including biological applications. The ability of this polymer to link with other materials, such as carbon nanotubes, is also under consideration.

## 2. Synthesis and Structure Characteristics

PAA, often known as carbomer, is an acrylic acid (AA) polymer with a carboxylic group (–COOH) on each monomer unit end is connected to the vinyl group. For its numerous carboxyl groups, poly (acrylic acid), a thermoplastic polymer, has substantial bioavailability and thus can be employed as a surface modification for biological nanomaterials [40].

Acrylic acid (propanoic acid) is a monomer that may be crosslinked to establish hydrogels with a higher moisture content capacity. This could be used as a single or multi-component structure. Acrylic acid has the general formula of CH=CH–COOH, and the carboxylic acid groups make this monomer a weak acid. The proximity of carboxylic acid groups makes the hydrogel ionizable, which can help increase its ionic strength and pH sensitivity. Acrylic acid monomers are also combined with various polymers to create multiple types of hydrogels [41,42,43]. Acrylic acid polymerization can occur in an acid medium employing materials like chlorosulfonic acid and sulphuric acid. Furthermore, polymerization can occur in the presence of alkalis, iron salts, light, high temperatures, unpaired valence electrons in an atom, molecule, or ion, and peroxide mixtures [44].

When all carboxyl groups dissolve, PAA has a high negative charge density. Neutralization transforms the acrylic acid monomer to sodium acrylate monomer in dissolved sodium hydroxide. As a result, this polymer and poly sodium acrylate are among the most widely employed water-soluble anionic polyelectrolytes, such as dispersion compounds, superabsorbent polymers, and ion-exchange resin. As indicated in Figure 1, they are exclusively produced by radical polymerization of sodium (acrylic acid) or acrylic acid [45,46,47].

The mechanical characteristics of PAA are improved by crosslinking it. Polymeric moderators in PAA might even help increase their tensile strength [48]. Furthermore, the cross-linked PAA has an incredible amount of water absorption. Poly (acrylic acid) is non-weatherable and has good optical characteristics. Various organic and inorganic nanoparticles are used to make PAA composite materials. The addition of reinforcements to the PAA network substantially impacts the final’s shape, temperature resistance, mechanical characteristics, coating, and biomedical structure [49]. Combining the nanofiller with the PAA framework resulted in nanomaterials with significantly improved properties. PAA-derived nanocomposites have ushered in a slew of new technological frontiers. Adhesives, electronics, and biomedical applications all use PAA nanocomposites. Because of their unique properties and uses, PAA nanocomposites have sparked a lot of research attention. Further research should concentrate on the pattern connections in PAA-derived nanocomposites for application areas [50,51].

Because of its hydrophilic character, PAA offers higher sticking powers to objects as a stabilizing agent. The cross-linked polyacrylate can capture and hold one hundred times its weight in moisture. Propanoic acid with a high density of carboxylic groups may improve the hydrophilicity of the resultant nanocomposite or mix in biological applications. Due to its nontoxicity and absorption properties, PAA has high drug storage and delivery capabilities. The hydrophilic characteristic of PAA-based mixtures may diminish serum protein adsorption, which is desirable in some situations to reduce blood clotting. Furthermore, cross-linked PAA can be employed as a medicinal adhesive due to its excellent bonding strength. PAA has hardness and barrier qualities in film form, making it ideal for packaged food [48,52]. Poly (acrylic acid) varies its characteristics in response to changes in ionic strength and pH; for example, at pH < 4, precipitate occurs in aqueous solutions due to the carboxylate groups’ protonation making the polymer sparsely soluble in water [53].

Different radical polymerization processes, such as inverse emulsion, have been used to make poly (acrylic acid). The developing polymer chains are not soluble in the monomer during bulk polymerization, resulting in precipitation polymerization, which poses practical issues in mixing and heat transmission. Solution polymerization is a quick and easy way to get past these problems. To commence polymerization, redox initiation is a particularly effective method of creating free radicals under favorable circumstances. This technique is widely used in low-temperature emulsion polymerizations [54]. PAA can be made by hydrolysis of a narrowly dispersed poly (tert-butyl acrylate) (PtBA) sample, as demonstrated in Figure 2. Anionic or controlled radical polymerizations make the poly (tert-butyl acrylate) sample. Other types of PAA possessing block copolymers comprised of poly(n-butyl methacrylate), poly(methyl methacrylate), polystyrene (PS), and poly(2-vinyl pyridine), also polymers with a consistent branching pattern, such as star and comb polymers can be synthesized using this process [55].

Because of its functional groups, acrylic acid is one of the monomers that can polymerize through radical polymerization. As a result, the straight production of acrylic acid by managed radical polymerization has recently become prevalent. Since AA can interact with metal, atom transfer radical polymerization (ATRP) of PAA was complex [56]. The PAA homopolymers were first produced via direct polymerization of AA in dioxane at 120°C using nitroxide mediated polymerization (NMP) [57]. The most effective approach for targeted polymerization of AA was reversible addition-fragmentation transfer polymerization. The first straight polymerization of AA was in dimethylformamide (DMF) at low conversion. In controlled radical polymerization of AA in protic conditions, several reversible additive transfer polymerization (RAFT) compounds were examined phenoxyxanthates, and dibenzyl dithiocarbonates were shown to be the most appropriate RAFT products [58]. Side reactions in the radical polymerization of acrylic monomers include chain transfer to solvent, which restricts chain transfer to polymer and high molecular weight, resulting in branching architectures [59].

In a relatively short processing time, cobalt porphyrin derivatives are used to mediate the controlled radical polymerization of AA to create PAA with a high molecular weight and minimal polydispersity [60]. Aside from the environmental benefit, the polymerization level of acrylic acid is much higher through water than in some other solvents. Loiseau et al. recommended that polymerization in water using a water-soluble RAFT agent could decrease chain transfer to the solvent. Still, no test results were provided to back up this claim [61]. During γ-irradiation, RAFT agent-mediated acrylic acid polymerizations in the aqueous solution and bulk were done independently [62]. UV light was also used to RAFT polymerize acrylic acid in the liquid media at room temperature. Nevertheless, an extra device is necessary for either -irradiation or UV-irradiation. The architectures of the RAFT compounds will have a significant impact on the live feature of polymerization in a common RAFT reaction [63,64]. The optimum choice for a RAFT agent is dithiocarbonate, whose intermediate radical is less stable than dithioester, according to studies on the polymerization of polar monomers and AA by RAFT [65]. Ji et al. reported a great structure of PAA with a molecular weight using RAFT synthesis in the aqueous phase is a logical option [66].

It is worth mentioning that PAA is similar to glass solids without color at ambient temperature, with the former having a glass conduction temperature of 106 °C and the latter having a glass transition temperature of 230 °C, which cannot be measured immediately due to its high temperature but can be inferred using copolymer statistics. It is soluble in alcohols, methanamide or formamide, water and alkali water (pH of 8 or 9), and DMF. Theta solvents of PAA in which side chains react as perfect chains are 1,4-dioxane and 0.2 M aqueous hydrochloric acid [67,68]. It is also notable that theta solvents fall in between desirable and non-solvents. It is noteworthy to mention that theta solvents for poly (acrylic acid sodium) have been reported to be 1.5 M aqueous sodium bromide at 15 °C and 1.12 M aqueous sodium thiocyanate at 30 °C. Most PAA carboxyl groups do not react with water or 1,4-dioxane at neutral pH [69]. The chain design in 1,4-dioxane seems to be near the random coil because the main chain’s C–C connections are very flexible at ambient temperature. Once PAA is neutralized by sodium hydroxide, the rate of separation improves. Practically all of the sodium acetate sites for NaPAA disintegrate in water. NaPAA acts like a traditional polyelectrolyte in aqueous, with n negative charges on each 0.25 nm of the polymeric chains. The solution viscosity is much greater than the random coil solution with the same amount and polymer chain length. The long-range electrostatic repulsive force between two anionic types substantially stretches the polymer chains. This suggests that high-molar-mass NaPAA and partially neutralized PAA can be used as a thickener. It is employed in poultices to maintain drugs on the skin surface [70]. Several different methods are employed to examine and characterize PAA. Some are used in other fields, such as infrared spectroscopy, gas and liquid chromatography, spectrometry, and nuclear magnetic resonance; in contrast, others are used mainly in the branch of polymers, such as osmometry size-exclusion, field-flow fractionation, and light scattering [71].

## 3. Architectures of PAA

### 3.1. Nanofibers

Nanofibers (NFs) have gained a particular interest owing to their unique physical and structural properties, i.e., large surface area, increased porosity, small pore size and fiber diameter, increased flexibility during functionalization of the surface [72]. Additionally, these possess high liquid or air permeability and rapid internal surfaces and form strong hydrogen bonds. Garza et al. [73] fabricated the nanofibers of PAA by subjecting the solutions of PAA for centrifugal spinning with various concentrations (9 to 14%) and speeds (4000 to 8000× *g* rpm), revealing different architectures of PAA nanofibers. When centrifuged at 6000× *g* rpm with 12 wt %. The average diameter of nanofibers was found 1100 nm when 12 wt % of PAA solution was placed for centrifugally spun at 6000× *g* rpm while the size was decreased to 900 nm in the case of 8000 rpm with the same concentration suggested the evaporation of the solvent in the fast spin rate resulting stretching of the nanofibers. The smallest diameter was found in the case of 9 wt %, which suggested that lower concentration led to smaller fiber average diameter, Figure 3a–c. PAA/PVA (in various molar ratios) electrospun nanofibers were stabilized by thermal crosslinking at 140 °C. The average diameter was found 309 ± 87 nm, 340 ± 83 nm, 290 ± 61 nm, and 221 ± 45 nm, for the molar ratios of 19.81, 35.65, 55.45 and 83.17, respectively for PAA/PVA [74]. This decrease resulted from the increment of conductivity and decrement of viscosity with increasing the PAA ratio. Eventually, the membranes maintained their fiber-based morphology joining at their points of contact after water immersion unveiled the porous architecture of PAA [75].

### 3.2. Nanoparticles

Nanoparticles are materials with overall dimensions in the nanoscale, ie, under 200 nm. In recent years, these materials have emerged as important players in modern medicine, with clinical applications ranging from contrast agents in imaging to carriers for drug and gene delivery into tumors [76]. Nanoparticles of PAA have been extensively studied in biomedical applications such as drug delivery due to the unique capability to deliver drugs, genes, and proteins via the peroral route. The thiolated PAA nanoparticles were developed by Greindl et al. [77], whose architecture was covalently crosslinked via disulfide bonds. The cross-linkage of PAA with 2,2′-(ethylenedioxy)bis(ethylamine) (EDBEA) showed spherical morphology and 20–80 nm-sized nanoparticles [78], while the PAA-PS-Ag composite nanoparticles revealed spherical morphology with 3 ± 1.2 nm sized particles [79]. The exact morphology was obtained by Müller et al. [80] with a mean diameter <200 nm. The human fibrinogen binding kinetics depended on the size of negatively charged PAA/Au nanoparticles. The larger nanoparticles revealed binding with fibrinogen with a slower dissociation rate and increasing affinity. When the size of nanoparticles was 7 nm, the two nanoparticles were accommodated by each fibrinogen molecule, but when the size increased up to 10 nm, only one was adapted. The size increments up to 10–12 nm changes from one site to the two-site binding. The bound nanoparticles felt more coulombic repulsion when the diameter was increased. Due to the flexibility of both binding sites, one nanoparticle with a sufficient diameter (15–17 nm) was also found enough for the interaction of fibrinogen. Hence, more than 12 nm, multiple protein molecules were found, Figure 4i [81]. PAA-coated iron oxide nanoparticles showed two types of molar mass (1800 and 5000) due to the different architecture of PAA chains, which influenced the molar mass. The magnetic diameter of these nanoparticles was found in between 7.3 to 11.9 nm [82]. The architecture of nanoparticles of PAA-chitosan (CS) was dependent on the synthesis and pH of the synthetic medium. The nanoparticles at 4.5 pH (acetic buffer solution) revealed consistent and solid spherical particles unveiling PAA-CS nanoparticles’ matrix structure. PH 7.4 showed a dense core bounded by a fuzzy and diffuse coating 4 (ii). This architecture was due to ionic interaction between negatively charged PAA and positively charged CS. The different preparation processes of PAA-CS nanoparticles influenced the architecture of nanoparticles. When PAA was dropped into a solution of CS, the generation of PAA core occurred, and a membrane was formed on the PAA core surface resulting in a dark shell and soft-core spherical nanoparticles. On the other hand, When the CS solution was dropped in the PAA solution, the core of CS and membrane of PAA-CS were formed. There were no cavities formed in PAA-CS because of the not swelling of CS in acidic medium, Figure 4iii. These all structures were created due to the construction of samples, conditions of staining, etc. [83]. The PAA magnetic nanoparticles possessed uniform particles morphology with a 9.2 ± 2.6 nm average diameter while a hydrodynamic diameter of 246 ± 11 nm (*n* = 3) was measured by the dynamic light scattering (DLS) measurements [84]. The other magnetic nanoparticles of PAA had a 10 nm size and were semispherical in shape [85]. PAA-coated iron oxide nanoparticles revealed a 10.1 ± 2.4 nm mean particle size. These were stable in water, and variation in pH or enhancement in ionic strength resulted in aggregation of these nanoparticles in water [86].

### 3.3. Nanocapsules

Nanocapsules have been the most extensively studied for functional compounds delivery [87]. Nanocapsule possesses a large inner cavity which helps in the high loading of drugs and sustained release of drugs due to its capsule-like structure [88]. The hollow tailor-made 100 nm nanocapsules of PAA/CS were fabricated for antibiotic therapy by Belbekhouche et al. [89]. Nanocapsules of PAA-N-isopropylacrylamide (PNIPAm) hydrogel are presented in Figure 5i suggests the polymerization and crosslinking of PAA with PNIPAm to form the nanocapsule architecture and unveiled the round shape morphology (135 nm) for PAA-hydroxypropylcellulose (HPC) template particles. At the same time, the figure was seen in the 1st step above. After crosslinking with PNIPAm, the core (dark) shell (dusky) structure was seen, increasing the size to 230 nm. Even after removing the template, the particles maintained the spherical morphology with a larger inner cavity and thin shell having 50 ± 12.5 nm thickness, Figure 5ii [90]. The Nanosphere of PAA/BSA showed the 80 nm diameter while the nanocapsules revealed the 300–500 nm. These were synthesized using in situ polymerization, swelling, and re-aggregation. The interior diameter was found 100–200 nm, and glutaraldehyde (GA) cross-linked PAA/BSA nanospheres increased the stability. After absorbing the water molecules into PAA/BSA/GA also, nanocapsules were also formed. Microspheres presented the porous shape, and the hollows were small in nanocapsules that suggested that the architecture of PAA/BSA was fixed by cross-linking agents and reduced its flexibility, Figure 5iii [91]. Nanocapsules of PAA-*b*-PAN di-blocks were prepared using PAA macroinitiators using RAFT polymerization. The same architecture revealed monodisperse and spherical with 30~35 nm hydrodynamic diameter analyzed by dynamic light scattering and TEM. The aggregation of these nanocapsules was occurred by π-π interaction of the graphite layers [92]. In situ acrylic acid polymerization was done to obtain liposome nanocapsules coated with PAA with a mean diameter of 123 ± 21 nm [93]. The core-shell structure for copolymers of PAA was seen in the nanocapsules of PAA with an average diameter of 70 nm and 10 nm of shell thickness [94].

### 3.4. Other Structures

Poly (acrylic acid-b-isoprene) cross-linked micelle structures were synthesized using calcium phosphate coating (20 nm thickness) having 60 ± 9 nm mean diameter, revealing the mineralization near or at surface regions of PAA. Additionally, nanocages were also formed. These hybrid materials were found stable for numerous months in water. Even though it aggregated and mineralized with time, there was no change seen in crystallization and diameter even after eight months (Figure 6a,b) [95]. Spherical microspheres of PAA/PVA of sequential interpenetrating network crosslinked with GA were obtained via SEM analysis. Figure 6c suggested spherical morphology without any agglomerations. A smooth microspheres surface was obtained with no pores, and some particles were covered with polymeric debris, Figure 6d [96]

## 4. Bio-Conjugation with Other Materials

Surface modification has been considered an effective method of improving material performance, modulating their properties, and extending their applications [97]. The layer-by-layer assembly deposits many coating layers on the surface, and self-assembled monolayers need a match between surface and sorbate chemistry are the two dominant strategies in surface modification chemistry. After that, the coating polymers are considered a surface modification for various applications. Among these polymers, PAA, because of its outstanding surface adherent properties, induces functional groups and biocompatibility and has been extensively used as the coating agent [98]. Various surfaces containing metal oxides, gold nanoparticles, metal-organic framework, silica, and carbon-based materials (carbon nanotubes, graphene) were coated using this polymer. The PAA films coated onto the surface of nanomaterials can improve their stability and solubility and facilitate functionalization of them to make intelligent materials. PAA film may theoretically add anticancer medicines and contrast agents to the surfaces.

### 4.1. Metal Oxides

Fe_3_O_4_ nanoparticles (NPs) are many metal oxides widely used as coating agents on surfaces. The PAA-coated Fe_3_O_4_ NPs have an inherent magnetic feature that allows them to be collected using an external magnetic field [99]. These nanomaterials have magnetically assisted therapies, and MRI uses in nanomedicine. Yunn-Hwa Ma et al. used a coprecipitation approach to make Fe_3_O_4_ NPs, which they subsequently modified with a PAA film and recombinant tissue plasminogen activator (rtPA). The Fe_3_O_4_@PAA NPs were used for the targeted delivery of rtPA by using an external magnetic field. The findings demonstrated that using a magnetic field could improve nanoparticle accumulation in tumor tissue [100]. Diana Couth et al. constructed Fe3O4@PAA NPs ranging from 8–11 nm in a separate study. The in-vitro effect of Fe_3_O_4_ NPs coated with PAA and bare was evaluated on the induction of six cytokines. The findings displayed that both polymer-coated and bare NPs could induce all of the cytokines examined [101].

Shuo-Li Sun et al. synthesized Fe_3_O_4_ NPs coated with PAA and then imbedded with polyethylenimine. At the end of production, the fabricated system was used for delivering plasmid DNA by using an external magnetic field. The results demonstrated that magnetoreception efficiency in HEK 293T and U87 cells increased in an external magnetic field [102].

A new aluminum hydroxide nanostructures embedded with PAA coated Fe_3_O_4_ NPs were developed and studied as dual MRI/positron emission tomography (PET) contrast agents for cell imaging by Manuel Antonio González-Gómez et al. [103].

Zhaoqiang Zhang et al. synthesized a biocompatible and superparamagnetic hollow mesoporous nanoparticle based on Fe_3_O_4_ NPs with the ability of magnetic targeting. Then the nanoparticles were coated with PAA, which can load bleomycin (BLM) through bonding with PAA in the mesoporous structure (Figure 7A–C). The results demonstrated that designed systems could effectively load drug and release it sustainably. The therapeutic efficacy of hollow magnetic NPs was much higher than free drugs [104].

Daniela Rodrigues et al. studied in-vivo biodistribution of PAA coated Fe_3_O_4_ NPs. In their study, after 24 h of intravenous administration of NPs, the expression of iron in mouse liver and spleen tissues was measured using histochemistry. According to the findings, iron deposition was found in macrophages from both organs [105].

Arkaban et al. designed a theranostic system based on CoFe_2_O_4_ coated with PAA and conjugated with folic acid (FOA) and doxorubicin (Dox) for theranostic intentions. This combination displayed increased diagnostic and therapeutic efficiency [106]. The theranostic system showed a size around 43 nm in diameter, while the thickness of PAA layers was about 12 nm. Finally, the theranostic system was used to treat cancer breast cells based on its chemotherapy influence; compared to non-targeting systems and free Dox, this method has better chemotherapeutic properties.

Other metal oxides were also covered with PAA to create usable nanoplatforms. T1-weighted MRI agents such as Mn oxides can be employed for imaging-guided therapy. Marzieh Samiei Foroushani et al. presented a multifunctional theranostic structure constructed of manganese oxide (Mn_3_O_4_) NPs, covered with PAA, and loaded with methotrexate targeting agent and anticancer drug. PAA, a pH-sensitive agent for loading and delivery of MTX, can improve MTX accumulation in tumor sites [107].

Arkaban et al. fabricated a nanocomposite system, including Au NPs, coated sequentially by MnCO_3_/Mn_3_O_4_ and PAA. Then, the PAA-immobilized NPs were imbedded with FOA (as targeting agent), Dox (anticancer drug), and propidium iodide (fluorescence imaging agent). The fabricated nanosystem displayed good encapsulated drug and encapsulating efficiency and increased ability for catching of 4T1 cancer cells when compared with non-targeted system and free Dox [108].

The manganese dioxide nanoparticles (MnO_2_) coated with PAA were synthesized. Their influence on lung cancer cells with or without gefitinib was reported by Me Hyeon Cho et al. In MR imaging, MnO_2_ NPs exhibited glutathione (GSH)-responsive dissolution and subsequent enhancement. Moreover, the therapeutic data demonstrated upon using X-ray irradiation, the therapeutic efficiency of MnO_2_ on lung cancer cells was considerably enhanced [109].

Khan et al. discussed the possible interactions between materials such as poly(xyloglucan-co-methacrylic acid), hydroxyapatite, and silica [110]. Figure 8 depicts a possible interaction between metal oxide and PAA.

### 4.2. Gold Nanostructures

Gold nanostructures have been considered in various fields because of their excellent physicochemical features, such as nontoxicity, suitable biocompatibility, simple methods for preparation, and excellent optical properties. Different gold structures have been fabricated, containing nanospheres, nanoclusters, and nanorods. The surface modification of gold nanomaterials with PAA for nanomedical applications is the focus of this paper.

Su Pan et al. synthesized PAA-coated gold nanorods (GNR@PAA). First, Cetyltrimethylammonium bromide-coated gold nanorods were synthesized by a seed-mediated procedure. Second, the nanorods were further coated with a PAA film. Finally, they discovered that hyperthermia therapy was more effective when GNRs@PAA was combined with a laser. They investigated the process of GNRs@PAA radiation treatments. They found that disrupted cell membranes and DNA integration cause cell apoptosis and death, with the cell apoptosis rate boosted by in vitro photothermal therapy [111].

In a separate Guilan Li et al. studied a standard method for fabricating gold nanorod@polyacrylic acid/calcium phosphate (AuNR@-PAA/CaP) core-shell NPs containing PAA/CaP shell and an Au rod as the core. They showed that AuNR@PAA/CaP core-shell NPs had a high drug encapsulating ability (1 mg Dox/mg NPs), excellent photothermal feature (26%), and pH/near-infrared dual-sensitive behavior. Because the CaP shell is destroyed at low pH values, releasing Dox increases. NIR irradiation of the Dox loaded in the AuNR@PAA/CaP core-shell NPs was released. AuNR@PAA/CaP core-shell NPs have also been effectively used in synergistically dual mode X-ray computed tomography/photoacoustic imaging as well as chemo-photothermal cancer treatment [112].

Gold nanostars (GNs) were coated using PAA sheets in addition to nanorods to create usable nanoplatforms. Spherical gold nanocrystals covered with PAA/mesoporous silica shell NPs (AuNC@PAA/mSiO_2_ NPs) with having accumulation enhanced fluorescence (AIF) properties were described by Xiaotong Wu et al. The manufactured NPs have been used as therapeutic and diagnostic agents for liver cancer chemo-therapy and synergistic fluorescence/X-ray computed tomography imaging. Surprisingly, the produced NPs had many AIF characteristics (equivalent to 4.2 times the individual AuNCs) and high drug loading and pH-responsive drug release [113].

Chixia Tian et al. described an MRI/CT bimodal imaging agent constructed of Gd-MOF and gold nanoparticles (AuNPs). PAA was employed to bridge Gd-MOF NPs and AuNPs, fabricating hybrid Gd-MOF/AuNPs. The obtained hybrid NPs were then estimated in dual imaging (MRI and CT). The findings demonstrated great relativity in MRI and CT imaging [114].

Gold nanoparticles can be integrated with PAA for several applications. Rezvani et al. created various core-shell NPs with AuNPs as the core and stimuli-sensitive polymers such as PAA, poly(N,N methylene bis(acrylamide))(PMBA), poly(methacrylic acid) (PMAA), poly(2-hydroxyethylmethacrylate)(PHEMA), and poly(N-isopropylacrylamide) (PNIPAAm), as shell [115]. According to TEM pictures, all core-shell NPs were smaller than 100 nm. The loading efficiency of systems was studied. Researchers observed that Au-PMAA and Au-PAA NPs had a high amount of drug loading because of effective interaction between the carboxyl groups of Dox and polymer. In addition, drug release was significantly increased under NIR light.

Zhou J. Deng et al. were synthesized Au NPs coated with PAA with sizes 7–22 nm and examined their interactions with fibrinogen protein. They monitored the binding kinetics of human fibrinogen to negative PAA-coated Au NPs and understood that the bigger NPs bound fibrinogen with high attraction and a gentler disconnection rate. While each fibrinogen molecule could collect two 7 nm NPs, only one fibrinogen molecule could aggregate 7 nm NPs. Several fibrinogen molecules were connected by NPs bigger than 12 nm. In any case, fibrinogen produced aggregation of the larger particles in the presence of additional NPs, which might connect more than one protein particle. This is comparable with fibrinogen interparticle bridging. Overall, the findings indicate that appropriate modifications in NP size can alter protein binding both on the NPs’ surface and within the protein corona [81].

Chunyuan Song et al. designed a nanocarrier with excellent loading efficiency and pH-responsive release behavior based on a flower-like Au NPs. the surface-enhanced Raman scattering active floral nanoparticles with a high surface area were synthesized and subsequently modified with thiolated-PAA (PAA-SH) for efficient pH-dependent loading and release of Dox as an anti-cancer medication [116].

### 4.3. Silica Nanoparticles

Lu Li et al. suggested a synthetic method for the fabrication of Fe_3_O_4_@mSiO_2_@PAA nanoclusters (NCs) and studied their applications in MRI and as a pH-sensitive DDS. First, they converted the oleic acid-capped Fe_3_O_4_ NPs to Fe_3_O_4_ coated with CTAB and then the formed fluorescein isothiocyanate (FITC)-labeled fluorescent mSiO_2_ shells on CTAB-Fe_3_O_4_ NPs. Then, the -synthesized Fe_3_O_4_@SiO_2_ NPs modified with the PAA shells. Finally, the obtained NCs were applied as DDS and fluorescent labels for imaging and therapy intentions. The results revealed that the fabricated system could load a lot of drugs and release them in a pH-dependent manner. Moreover, in-vitro tests verified that the NCs are biocompatible, and the Dox-loaded NCs had a high cytotoxic effect on cancer cells [117]. Also, Zhen Xia et al. designed a DDS constructed of CaF_2_:Yb,Er NPs coated with SiO_2_ nanofibers (CaF_2_:Yb,Er@SiO_2_) for Dox delivery. The results showed importing PAA on silica nanofiber can improve the loading ability of the fiber. Also, PAA-coated CaF_2_:Yb,Er@SiO_2_ nanofibers revealed light and pH-dependent release. Dox release rate and in-vitro anti-cancer efficacy were improved by irradiation with a NIR (808 nm) laser [118].

### 4.4. Metal-Organic Frameworks (MOFs)

MOFs fabricated of metal ions and organic linkers. These porous crystalline materials are used in various fields because of having an excellent surface area, stability, porosity, and biocompatibility [119]. MOFs are frequently utilized as nanocarriers because of having good loading efficacy and sustained drug release properties [120].

Tran et al. constructed a nanostructure based on ZIF8 with high encapsulation ability for Dox delivery (Dox-loaded ZIF8). The Dox-loaded ZIF8 was modified with PAA (ZIF8–Dox@PAA) (Figure 9). Finally, they have studied the release manner of Dox at different pH and the influence of various factors on its release [121].

In another study, Amin Bazzazzadeh et al. fabricated the magnetic MIL-53 particles coated with PAA grafted-chitosan/polyurethane core-shell NPs for delivery of temozolomide (TMZ) and paclitaxel (PTX) toward U-87 MG glioblastoma cells [120].

### 4.5. Carbon Nanomaterials

Ming Xu et al. determined the toxicity effect of GO. They found from in-vitro experiments that pristine GO could hurt cell functions and cell membrane integrity. To increase the biocompatibility of GO, they were modified with poly(acrylamide), poly(ethylene glycol), and PAA. GO coated with PAA revealed the most biocompatibility [40].

The multi-walled carbon nanotubes (MWCNT) combined with iron oxide NPs because of their unique properties (nontoxicity, magnetic features) have been considered drug carriers. The low circulation time in biological fluids is one of the most significant drawbacks of these NPs. To overcome this problem, Bardajee et al. coated MWCNT with PAA and swelling kinetics and their capability to load and release tetracycline hydrochloride in various conditions studied [122].

Yunping Chen et al. synthesized graphene nanosheets (GNSs) by direct current arc discharge and functionalized and loaded them with hydrophilic PAA and Dox, respectively (GNSs-PAA(Dox)_load_). Results showed that PAA(10 wt%)-GNSs significantly improve the solubility of GNSs in aqueous solution and have a high loading efficiency of 2.404 mg/mg at the concentration of 0.36 mg/mL of Dox. Also, the PPA-GNSs showed acceptable pH-sensitivity [123].

At two distinct pH values, C. Sgarlata et al. investigated the physisorption process, and hydration behavior of gemcitabine (GEM) PAA GO. The energy of physisorption and the hydration shell around the complex is affected by the varied ionization of pH-sensitive groups. A decrease aids the physisorption process between PAA and GO in negative charge density, occurring at acidic pH. However, a modest interaction between GEM and PAA is found at the same pH setting. Compared to unionized GO and bulk water, the radial distribution function (RDF) shows that carboxylate oxygens of PAA and alkoxide oxygens of GO significantly attracted dipolar water molecules, affecting the hydration shell around the complex [124].

### 4.6. Other Metals

Adibehalsadat Ghazanfari et al. prepared five types of PAA-covered small metal oxide NPs with an average size of 2.3, 1.7, 1.5, 1.8, and 1.9 nm, respectively (Bi_2_O_3_, Yb_2_O_3_, NaTaO_3_, Dy_2_O_3_, and Gd_2_O_3_) and characterized their X-ray attenuation features, and accomplished in-vivo CT imaging using of the samples. Results showed that all NPs have outstanding colloidal stability and biocompatibility, and X-ray attenuation powers are more significant than those obtained for commercial iodine contrast. They extracted X-ray attenuation efficacies 11.7, 6.8, 10.3, 6.1, and 5.9 HU/mM for PAA-coated ultrasmall Bi_2_O_3_, Yb_2_O_3_, NaTaO_3_, Dy_2_O_3_, and Gd_2_O_3_ NPs, respectively. Also. They investigated the NPs as CT contrast agents in CT images in the mouse organs [125].

Xuekun Jia et al. successfully synthesized PAA-modified NaYF4:Yb, Er NPs (PAA-UCNPs) with the dual drug carrier and imaging capabilities. They used the PAA as a pH-responsive system to encapsulate drugs via electrostatic interaction. The drug encapsulation efficacy of the PAA-UCNPs was examined using Dox to assess their potential as a nanocarrier system. The loading and release of Dox loaded on PAA-UCNPs depended on varying pH. While a low amount of Dox was released at a weak alkaline medium, an increased release was observed in an acidic medium. The in vitro cytotoxicity test indicated that the PAA-UCNPs loaded with Dox were cytotoxic to HeLa cells [126].

Yufei Ma et al. produced and employed NaYF4:Yb^3+^,Er^3+^ NPs (UCNPs) to monitor rabbits’ bone tissue mesenchymal cells (MSCs). To increase biocompatibility and cellular uptake of NPs, they coated the UCNPs with a negative polymer PAA and a favorable poly(allylamine hydrochloride) (PAH-PAA-UCNPs). In terms of ALP activity, osteogenic protein expressions, cell viability, and the generation of mineralized nodules, no significant difference was identified between UCNPs-free MSCs and MSCs labeled with UCNPs (concentration range of 0–50 g/mL) (concentration range of 0–50 g/mL) [127].

Yan Ma et al. produced Co_0.85_Se NPs (PAA-Co_0.85_Se NPs) using an ambient aqueous precipitating technique for dual photothermal-chemotherapy of malignancies. PAA Co_0.85_Se NPs with outstanding photothermal conversion efficiency (45.2%), low cytotoxicity, significant near-infrared (NIR) light absorption, and ultrasmall size (8.2 nm) were produced. Dox encapsulated on PAA-Co_0.85_Se NPs with a loading efficiency of 8.3%, which showed a pH-sensitive release behavior because of the protonation of carboxyl groups in PAA molecules and amino groups in Dox. Also, they studied the cytotoxic effect of PAA-Co_0.85_Se-Dox NPs on HeLa cells. Irradiation with a near-infrared laser had a significant synergistic cell killing impact and increased treatment efficacy [128].

Hanzhu Shi et al. synthesized PAA/(CaCO_3_) NPs using a primary and new procedure. PAA/CaCO_3_ NPs were not only substantially more effective at loading Dox (1.18 g of Dox per g of NPs), but they also had a pH-sensitive characteristic. In vivo tests revealed that Dox-loaded PAA/CaCO_3_ NPs have a considerable anticancer impact with no noticeable adverse effects [129].

Kai Zhang et al. designed a drug carrier based on PAA-adorned three-dimensional (3D) MoS_2_ NPs (PAA-MoS_2_ NPs) that respond to NIR laser irradiation for the treatment of hypertension utilizing atenolol (ATE). The drug encapsulation efficacy and photothermal converting effect of PAA-coated MoS_2_ NPs were also investigated. The PAA-MoS_2_ NPs had a high drug-loading ability of 54.99 percent and a high photothermal efficiency. Further, they have explored the controlled release capacity of the PAA-MoS_2_ NPs using in-vitro drug release and skin-penetration studies. In the laser-stimulated group, drug release was 44.72 percent, and skin permeability was improved by a factor of 1.85 [130]. PAA-based core-shell nanostructures on various substrates are summarized in Table 1.

## 5. Biomedical Applications

Nanotechnology is the manipulation of matter on a near-atomic scale to produce new structures, materials and devices. The technology promises scientific advancement in many sectors such as medicine, consumer products, energy, materials and manufacturing [172,173,174,175]. New developments in science, bioinformatics and nanotechnology have significant impact on human health and life [176,177,178,179]. For example, bioinformatics is interdisciplinary fields, which harnesses computer science, mathematics, physics, and biology [180,181,182,183]. In other hand, PAA is employed in adhesives, coatings, homes, packaging, pharmacology, and other medical and biological industries [25].

### 5.1. Bio-Sensing

Hydrogels with excellent mechanical strength, rapid recovery, and shape memory capabilities based on PAA could open up new possibilities for various biomedical applications [184]. Carboxymethyl xylan-g-poly(acrylic acid) was designed to have strong compression strength, elongation, and elasticity, as well as shape memory capabilities activated by Fe^3+^ [185]

Endotoxins, commonly known as lipopolysaccharides (LPS), are infections produced from gram-negative bacteria’s outer membrane and cause serious harm to humans. LPS detection that is sensitive and selective is in high demand, notably in medical supplies, pharmaceuticals, and food. MoS_2_-PAA nanocomposite loaded with Au NPs was fabricated for the detection of LPS. Firstly, MoS_2_ nanosheets were prepared using sonication-assisted exfoliation of bulk MoS_2_ with PAA, and then it was immobilized using thiol terminated LPS binding aptamers, which were combined with Au nanoparticles [186].

The plasma polymerization approach was utilized to prepare electrode material consisting of hollow TiO_2_ spheres and PAA for detecting lysozyme. According to electrochemical impedance spectroscopy data, the produced TiO_2_ at PAA aptasensor has a very sensitive detection ability toward lysozyme; the proposed aptasensor has a detection limit of 0.015 ngmL^−1^ in the range of 0.05–100 ngmL^−1^. The film showed good selectivity for lysozyme in the medium containing interfering proteins such as immunoglobulin E, bovine serum albumin, and thrombin [187].

Nanospheres as immunoprobes were produced using chitosan-poly(acrylic acid) nanospheres doped with copper, cadmium, lead, and zinc ions to detect electrochemical signals and react with glutaraldehyde to immobilize various tagged antibodies [188]. Hydrogen peroxide with an excellent detection limit of 0.5 μM using Met-hemoglobin was developed as an electrochemical biosensor [189].

Microgels’ visible color and distinctive spectral features based on poly (acrylic acid-co-N-isopropylacrylamide) microgel-based have been demonstrated, and both depend on solution temperature and pH. Its sensitivity will be further exploited [190].

PAA brushes having carboxyl groups should be adaptable enough to allow for a wide range of chemical modifications, including the attachment of bioactive species that can be used as biosensor detecting probes. According to this research, PAA brushes with a predetermined graft density have shown to be a suitable precursor layer for biosensing applications [191].

Moreover, poly(tert-butyl acrylate) brushes were synthesized using the polymerization of tert-butyl acrylate. The tert-butyl groups from the poly(tert-butyl acrylate) bushes were removed by acid hydrolysis, yielding PAA brushes. The PAA brushes’ carboxyl group density can be adjusted based on chain length or molecular weight. The carboxyl groups of PAA brushes were tested to immobilize biotin [192].

Functionalized polymer based on polystyrene core and PAA shell nanospheres with cadmium ions was used to detect human IgG. The carboxyl groups of PAA shells were used to chelate cadmium ions and then conjugate them with antibody (Ab2) to generate metal ions labeled bioconjugates that were used as the label in immunoassays. In the future, this method is projected to be used extensively in protein diagnostics and bioanalysis [193].

PAA- multiwalled carbon nanotubes are a hydrophilic composite poly(acrylic acid)-a wrapped complex demonstrating remarkable stability in basic and acidic pH conditions. The complex also shows strong resistance to moderate ionic strengths [194].

### 5.2. Bio-Imaging

Bioimaging is a critical diagnostic technique for studying and visualizing biological events in cells and medicine [195]. Chen et al. proposed a simple production approach for synthesizing nanoparticles based on PAA using atom transfer radical polymerization (ATRP). When activated by a 980 nm near-infrared laser, the nanoparticles disperse efficiently in water and emit a strong green light, indicating that they could be helpful for luminous bioimaging [196].

Uniform Nd^3+^-doped LuVO_4_ nanophosphors were produced and surface-coated with PAA; these nanoparticles are in a colloidal range in physiological pH range and have excellent stability to survive in the cell. Because of these features, these may be used as a bimodal probe for X-ray computed tomography and NIR luminous bioimaging [197] CdSe/Cu quantum dot conjugates with biocompatible polyacrylic acid functionalization were produced for bio labeling, bioimaging, and biomolecule detection applications [198].

Polycrystalline and crystallized into a hexagonal shape of PAA-Eu-NaGdF_4_ nanospheres were fabricated by Nunez et al. Their size could be changed in the 60–95 nm range, varying the amount of PAA applied. When activated with UV light, these nanoparticles showed red luminescence, and the smaller nanospheres showed tremendous promise to MRI [199].

To functionalize Bi- and Eu-doped NPs based on rare earth vanadates (MVO_4_, M = Gd, Y), poly(allylamine hydrochloride) and PAA were used. The cytotoxicity, colloidal stability in various biologically relevant buffer media, absorption by HeLa cells, and low pH degradability also proved suitable for bioimaging and biosensing applications [200].

Optical bioimaging has emerged as a vital technique for detecting diseases with great sensitivity. NaYF4:Gd/Yb/Er nanorods modified by PAA were fabricated by Xue et al. with increased NIR IIb emission for bioimaging applications. Without a craniotomy, non-invasive optical brain vascular bioimaging with excellent spatial (43.65 m) and temporal resolution is acquired through the scalp and skull [201]. NaLuF4: Gd/Nd nanorods modified by PAA were studied for high sensitivity in vivo optical imaging and NIR-II bioimaging-guided small tumor diagnosis. The NIR-II emission of the NaLuF4: Gd host can be easily changed by doping Nd^3+^, resulting in a potential emission with strong photo-stability centered at 1056 nm and 1328 nm [202].

### 5.3. Cancer Therapy

New developments in science, bioinformatics and nanotechnology have significant impact on human health and life [203,204,205,206,207]. Cancer is a group of diseases involving abnormal cell growth with the potential to invade or spread to other parts of the body [208,209,210,211,212,213]. In this light, a hybrid material based on PAA and mesoporous silica nanoparticles was used for drug delivery systems in various ways, including surface modification, doxorubicin hydrochloride loading, and PAA coating. Doxorubicin was used as a model guest molecule to study drug encapsulation and release behavior at various temperatures and pH levels. It has the advantages of being easy to make, having no cytotoxicity, and having a high drug loading capacity, all of which could be useful in anticancer therapy [214]. A nanosized polyacrylic acid-polyaniline copolymer with increased water solubility was developed and proved to be an excellent option for wound healing and cancer therapy, particularly in the treatment of HT29 [215].

By using norepinephrine-loaded PAA nanogels as angiotonics, Li et al. have developed an anticancer auxiliary delivery method. The auxiliary system significantly reduced nano-drug uptake in the liver by raising the liver blood flow rate. The blood perfusion quantity rose dramatically by about 200 percent after administration of the as-prepared norepinephrine-loaded PAA nanogels, as measured directly by ultrasonic imaging, showing a higher blood flow rate in the liver [216]. For paclitaxel targeted delivery and anticancer efficacy, Reddy et al. designed sodium alginate grafted poly (acrylic acid–co-acrylamide/cloisite-30B/silver nanoparticle hydrogel composites with different weight percentages of cloisite-30B clay [217].

Erlotinib (ETB) is a commonly prescribed medication for non-small-cell lung tumors. To severe toxicity in clinical applications and avoid drug resistance, pH-sensitive and redox-responsive nanocarriers were developed to encapsulate ETB. Poly (acrylic acid)-cystamine-oleic acid was produced by emulsification followed by solvent extraction and converted into ETB-loaded lipid nanoparticles for lung cancer treatment [218].

Based on the bionics concept, Stimuli-responsive polymer materials are a new class of intelligent materials that exhibit more significant changes in physicochemical properties when activated by small environmental stimuli, making them an excellent carrier platform for anticancer medication delivery. Hybrid block copolymers based PAA, poly(2-methacryloyloxyethyl ferrocenecarboxylate), and Fe_3_O_4_ with dual stimuli responsiveness were developed using a two-step sequential reversible addition-fragmentation chain transfer polymerization process. It was non-toxic, stable, and entrapped hydrophobic anticancer drugs, which were then delivered quickly in particular microenvironments, including acidic pH and high reactive oxygen species [219].

Because it improves efficacy while reducing adverse effects, oral chemotherapy is the preferred method for cancer treatment. Unfortunately, the oral bioavailability of most anticancer medicines was inadequate. Tian et al. reported a pH-triggered oral medication delivery system using a simple graft-onto technique to cap mesoporous silica SBA-15 with pH-responsive PAA. PAA-capped mesoporous SBA-15 had a high drug loading capacity (785.7 mg/g), was pH sensitive, and had good biocompatibility. This pH-activated oral drug delivery device could be helpful in the treatment of colon cancer and other disorders [220].

Using a graft-onto approach, a nano-carrier based on PAA as shell and mesoporous hydroxyapatite nanoparticles as the core was developed, with an anti-proliferative effect on cancer cells. By electrostatic interactions, the grafted PAA may significantly enhance the loading quantity of the medication doxorubicin hydrochloride (DOX) as a pH-responsive switch [221].

Yuan et al. developed a nanoassembled drug delivery platform for antitumor therapy based on host-guest relations between cyclodextrin (CD) modified poly(acrylic acid) (PCDAA) and paclitaxel (PTX). In H22 tumor-bearing mice, PCDAA-PTX NPs can successfully target the tumor site due to their improved permeability and retention effect. PCDAA-PTX NPs have better effectiveness in suppressing tumor formation in vivo than commercially available anticancer medicines [222].

A magnetic cisplatin-encapsulated nanocapsule with a cisplatin-PAA core in an amphiphilic polyvinyl alcohol/iron oxide nanoparticles shell is made double emulsion to give excellent loading efficiency and regulated drug release. A549 tumor-bearing mice showed antitumor effectiveness with minimal adverse effects [223]. Lee et al. studied the anticancer efficacy of nanoparticles made from poly(methyl methacrylate-co-acrylic acid), including cisplatin in vitro and in vivo [224].

### 5.4. Cancer Theranostic

Integrating multimodal imaging and therapeutic functionalities into a nanoplatform has been identified as a viable cancer treatment method. However, several obstacles remain, such as instability and the difficult synthesis process. Zhao prepared clearable MnCo_2_O_4_ nanodots modified with PAA as nanoagents for T_1_/T_2_ bimodal MRI imaging-guided PTT. The single MnCo_2_O_4_@PAA nanomaterials can be used as contrast agents for T_1_/T_2_ bimodal MRI, owing to their intrinsic magnetic ability and precise diagnostic information [225]. CoFe_2_O_4_@PAA-FA Doxorubicin (Dox)load NPs were used to create a multifunctional theranostic nanocomposite for multifunctional cancer therapy [106].

For both controlled drug delivery and diagnostic feature, an integrated nanocomposite system was developed that contained ZIF-8 and PAA (pH-sensitive agent), manganese oxide nanoparticles (tumor diagnostic agent), and methotrexate (therapeutic agent and tumor biomarker agent) [147].

In another study, the AuNPs@MnCO_3_/Mn_3_O_4_@PAA nanoplatform is built, consisting of Au NPs doubly coated with MnCO_3_/Mn_3_O_4_ polyacrylic acid. After that, folic acid is conjugated to the immobilized polyacrylic acid, then doxorubicin and propidium iodide are added (as fluorescence agents, targeting and therapeutic) [108].

The preparation of PAA-prussian blue-Au aggregate janus nanoparticles has been reported. This heterostructure also adds drug loading functionality, which can be used in computed tomography imaging-guided chemotherapy, and enhanced photothermal therapy promotes tumor inhibition [226].

Zhao et al. used a straightforward one-pot solvothermal approach to make a PAA-functionalized porous BiF_3_: Yb, Er nanocarrier. As a result, carboxyl-functionalized BiF3:Yb,Er is projected to be an excellent choice for temperature sensing and multifunctional theranostic nanoplatforms development [227]. A multifunctional core-shell contrast agent made of PAA/calcium phosphate (CaP) a shell and spherical Au nanoclusters assemblies as core have developed by Li et al. Furthermore, the doxorubicin-loaded nanoparticles may be used as synergetic pH-sensitive drug delivery vehicles in vivo for dual-modal fluorescence imaging and computed tomography-guided liver cancer treatment [228].

A repeatable and straightforward synthetic technique synthesizes PAA/mesoporous silica shell nanoparticles with gold nanoclusters aggregation and increased fluorescence characteristics. In vitro and in vivo, the as-prepared NPs were used as new theranostic agents for liver cancer chemotherapy and synergistic fluorescence/X-ray computed tomography imaging [113].

Wu et al. developed hybrid PAA-Fe_3_O_4_ nanogels that can be used for both drug delivery and magnetic resonance imaging. An in situ co-precipitation method encapsulated superparamagnetic Fe_3_O_4_ nanoparticles inside porous PAA nanogels with high drug loading release [140].

Nanobubbles have the potential to be novel theranostic systems for ultrasound, magnetic resonance imaging (MRI), combined high-intensity targeted ultrasound-triggered drug release, and magnetic targeting for the treatment of cancer. A single-step emulsion approach was used to make nanobubble-based dual contrast enhancement agents from thermosensitive F127, and PAA stabilized with superparamagnetic iron oxide nanoparticles and encapsulated with perfluorobutane. This nanobubble system’s combination of functions makes it a potent and practical new tool for achieving effective cancer treatment and in vivo tumor imaging [229].

### 5.5. Tissue Engineering

An electrospun nanofiber (NFs) have shown excellent biomedical application such as controlled drug release, wound dressing, and tissue engineering. It also could be a promising candidate for postoperative chemotherapy. Hajikhani et al. reported the encapsulating properties of nanofibers derived from electrospinning of copolymers of PAA, polylactic acid, cellulose acetate, and polyethylene oxide for controlled release lycopene [230]. Khajeh et al. reported using biocompatible nanofibers derived from electrospinning of PAA, poloxamer, and polyurethane for wound dressing. The MTT experiment demonstrated that the generated NFs were non-toxic to the cells. The cell adhesion investigation revealed that the developed NFs could be used as a platform for proliferation and cell adhesion [231].

Ghaffari-Bohlouli et al. created molecularly imprinted polymer nanoparticles from an electrospun blend of PAA and poly(L-lactide-D, L-lactide) as well as poly (2-hydroxyethyl methacrylate) to form nanofibers with an average diameter of 237 nm for bone tissue engineering applications [232].

Due to its remarkable properties, such as excellent biocompatibility, minimal frictional behavior, and high water content, poly(vinyl alcohol) hydrogel has been regarded as a suitable cartilage replacement material. However, PVA hydrogel’s lack of mechanical characteristics and cytocompatibility are two significant challenges to its use as a cartilage substitute. To counteract these issues, PAA has been added to the PVA hydrogel. PVA/PAA hydrogel offers comparable biocompatibility to pure PVA hydrogel and much better cell adherence. As cartilage tissue substitutes, these biocompatible composite hydrogels provide a lot of potential [233].

Various polymers and polymer-based materials, including PAA and its derivatives, have been extensively described in biomedical applications. Medical probes for analysis, biocidal actions against various diseases, hand-held water filters, surface coatings, and fibrous disinfectants are all made with these materials [234]. For example, Nurkeeva and colleagues examined the antibacterial activities of PAA and its derivatives as well as the production of PAA complexes with streptomycin sulfate [235]. Larsson and coworkers developed composite films based on biodegradable polyhydroxybutyrate (PHB) and PAA nanogels for bone-tissue engineering applications [236]. Based on the radiation-induced inter-and intramolecular cross-linking of the inter-polymer complex of PAA and polyacrylamide (PAAm), Ghorbaniazar investigated the formation of nano-sized polymeric gels. MTT assay was used to assess and prove the biocompatibility of nanogels [237].

### 5.6. Antimicrobial Applications

Antibiotic-resistant microorganisms have become a significant public health issue. Biofilms are the primary cause of hospital-acquired infections and illnesses [238,239]. Once developed, it is challenging to remove biofilms because they feature antimicrobial defense systems [240,241]. Antimicrobial surfaces must kill or repel germs before settling and forming a biofilm [242,243,244]. Gratzyl et al. used anionic polymerization with acid as a catalyst to prepare diblock copolymers based on PAA and poly(styrene) and PAA and poly(methyl methacrylate) [245]. Antibacterial activity against pathogenic microorganisms such as *S. aureus*, *E. coli*, and *P. aeruginosa* has been demonstrated. The bactericidal activity of diblock copolymers increased as the acrylic acid content increased.

Interactions between negatively charged membrane groups and negatively charged acidic components are believed to emerge via the formation of salt bridges between divalent counter ions and acids such as Mg^2+^ and Ca^2+^, which balance the charge of the membrane on the surface of bacteria [246]. Multivalent carboxylic acids are likely to form chelates with Mg^2+^ and Ca^2+,^ destabilizing the membrane and potentially leading to cell death.

The acidic ion-exchange capacity of PAA is due to acid dissociation, which makes several deprotonated carboxylic groups available in slightly acidic circumstances and is pH-dependent [247]. Since the amount of COO- in this range is significant and provides a high affinity for cations and a high negative charge density, the best ion-exchange ratios of PAA can be produced in the pH 4.5–6. As a result, the ion-exchange potential of the material, which is defined by the polymer’s PAA content, which defines the cation affinity and negative surface charge, is the major determinant of antibacterial action. Sethy et al. synthesized PAA/GO/Ag nanocomposites in aqueous environments using an in-situ polymerization technique. It has shown excellent antibacterial activity against pathogenic bacteria [248].

According to Gratzl et al., the presence of PAA in the copolymer and slightly acidic conditions are essential for the material’s antibacterial activity, but counter-ions significantly limit its efficacy. These findings lead us to believe that the bactericidal action of the copolymer is due to an ion-exchange effect [249]. The antimicrobial activities of polymeric composites based on PAA and zinc were investigated using in situ solvothermal methods against pathogenic bacteria such as *B. subtilis* and *E. coli* and fungi such as *S. cerevisiae* [250].

Shibraen et al. reported that interactions with polyacid groups resulted in doping copper, iron, and silver metal ions into polyelectrolyte multilayer matrix. Cationic guar gum /PAA nanofilms loaded with Ag^+^ displayed strong antibacterial activity and could be exploited as a coating material for medical devices [251].

Previously, Nie et al. described the high-temperature synthesis of oleate-capped iron oxide nanoparticles (OIONPs), followed by a ligand exchange reaction between sodium polyacrylate and OIONPs to prepare PAA capped iron oxide nanoparticles. It was discovered to have potent antibacterial properties against *E. coli* and *S. aureus* [252]. Xu et al. developed a programmed antibacterial and remineralization technique for treating dental cavities using alendronate-grafted PAA /zinc-substituted hydroxyapatite hybrid nanoneedles [253]. Compared to PAA alone, the antibacterial capabilities of silver/gum acacia/PAA nanocomposite hydrogels have greatly improved [254]. A one-step hydrothermal deposition approach was used for coating applications to synthesize a PAA/gentamicin sulfate/hydroxyapatite. The antibacterial activity against *S. aureus* was determined using the plate-counting method [255].

Electrospinning is used to produce random and aligned PAA/PVA nanofiber scaffolds treated with cold atmospheric plasma. It has shown reasonably good antibacterial activity against *E. coli* [256]. Dil et al. reported the development of a nanocomposite hydrogel based on PAA, gelatin, and nanosilver, which demonstrated great antibacterial activity against harmful bacteria such as *S. aureus* and *E. coli* [257].

## 6. Compatibility and Biodegradability

In the biomedical field, biocompatibility and biodegradability are the main features of any material to be used [258]. PAA is a superabsorbent water-soluble polymer and is extensively used in several applications such as tissue engineering, disposable diapers [259], release devices [260], membranes [261], toothpaste [262], ion exchange resins [263], etc. Compatibility of PAA/Starch blends was studied and revealed that the glycerol’s incorporation into the mixture was responsible for the enhancement in hydrogen bonding between starch and PAA. This stiffness was enhanced by increasing the content of starch. The PAA/Starch blends were found fully amorphous and partially miscible [264]. PAA was crosslinked with dextrin to synthesize the biocompatible crosslinked hydrogel of c-Dxt/PAA for the sustained release of ciprofloxacin and ornidazole. In the biodegradation studies, it was noticed that the mass of c-Dxt/PAA was degraded progressively, which was done was lysozyme by glycosidic bond’s enzymatic hydrolysis by the hexameric sugar ring binding sites of the lysozyme. The degradation rates were diminished between 7 to 21 days once the reduction in the suitable site occurred [265]. Cytotoxicity studies of coated liposomes with PAA nanocapsules suggested less toxicity for PAA coated than without coating, suggesting its more biocompatibility [93]. Cytotoxicity studies for PAA nanocapsules were performed with A549 cancer cells and revealed negligible cytotoxicity, demonstrating nanocapsules’ biocompatibility at all concentrations. Also, the IC_50_ values were found lower, unveiled the possibility of more accumulation of PAA nanocapsules in the cells and can be used in vivo studies. The authors also studied the biocompatibility in in-vivo and suggested an electrostatic interaction between carboxylic acid of PAA and CDDP that improved the release of CDDP. The blood circulation time of these nanocapsules was increased compared to free drugs, and a 10-fold increase in the carrier for tumor accumulation revealed higher in-vivo antitumor activity. Therefore, PAA-CDDP combination with magnetic targeting was a better choice without significant bodyweight loss because of the effect of magnetic targeting [223]. The degradability of the PAA/PU semi-interpenetrating polymer networks was studied equally in normal and accelerated conditions. When the hydrophilic semi-interpenetrating polymer networks fraction was increased, the degradability was also found to be increased in three steps: incubation, induction, and erosion stage [12].

## 7. Conclusions, Challenges

Polymers have played a vital part in improving drug carriers by offering the controllable release of the encapsulated agent in a steady dosage over extended durations and controllable delivery of both hydrophobic and hydrophilic drugs. On the other hand, polymers that may modify their properties in response to environmental variables have recently received much interest. These polymers are stimuli-responsive materials because their functional groups make them extremely sensitive to their surroundings. PAA and its nanoconjugates could be regarded as stimuli-responsive platforms that make them ideal for drug delivery and antimicrobial applications.

The being safe of PAA-coated materials for biomedical applications should be more scientifically studied. Although PAA has admirable biocompatibility, the toxicity of these NPs for biological organisms should be approved in cell and animal experiments. Additionally, the time and mechanism required for the PAA drug delivery platform for a specific cellular compartment or tissue should be fully addressed.

## 8. Future Prospective

Nowadays, the primary objectives in this sector are to allow and promote more research activities aimed at developing competitive and translatable products. With this in mind, an open-minded and multidisciplinary approach may lead to the speedy and effective translation of developing PAA drug delivery systems shortly. PAA and its derived nanoparticles can be used as carrier materials for nano delivery systems and have many biomedical applications, such as drug delivery, vaccine delivery, antibacterial agent, and wound healing. Also, researchers should conduct in-depth studies on new usages for chitosan and also find out more about human-related effects through animal experiments. PAA and its derived nanoparticles will draw more and more attention and will have unlimited application prospects. Of course, merging biological and synthetic viewpoints will give a new perspective for the development of more effective polymeric nanoplatforms.

## Figures and Tables

**Figure 1 polymers-14-01259-f001:**
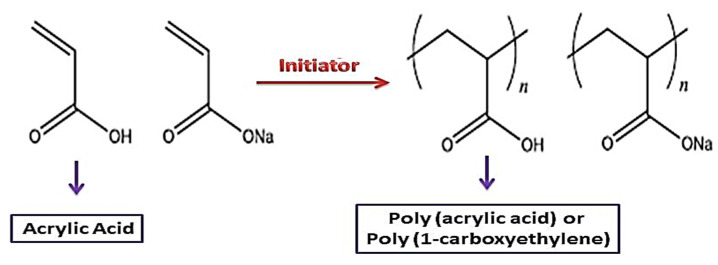
Synthesis and structure for poly (sodium acrylate) (NaPAA) and PAA.

**Figure 2 polymers-14-01259-f002:**
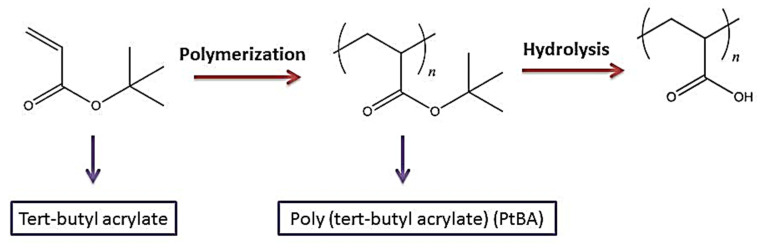
Synthesis for poly (sodium acrylate).

**Figure 3 polymers-14-01259-f003:**
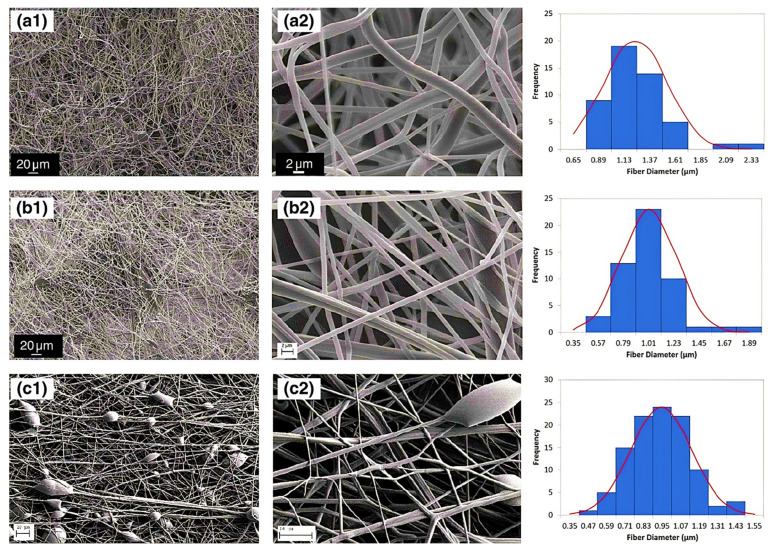
Diameter distribution and SEM micrographs of nanofibers of PAA (**a**) concentration of 12 wt% PAA and speed of 6000 rpm, (**b**) concentration of 12 wt% PAA concentration and speed of 8000 rpm, and (**c**) concentration of 9 wt% PAA concentration and speed of 4000 rpm. The SEM micrographs were taken at various magnifications: (**a1**,**b1**) 600×, (**c1**) 300×, (**a2**,**b2**) 7000×, and (**c2**) 15,000×. (Reprinted from Ref. [73] with permission).

**Figure 4 polymers-14-01259-f004:**
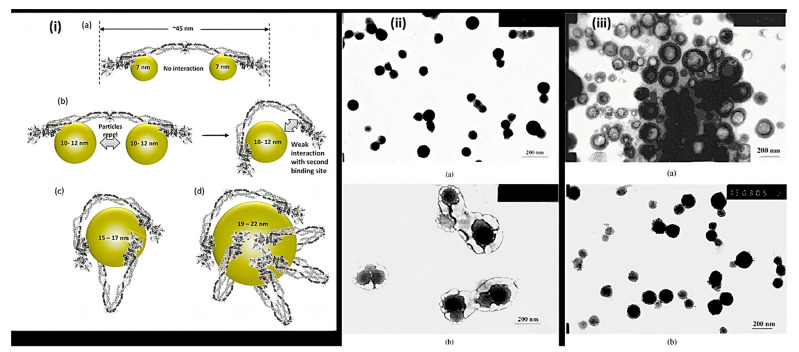
Representation of (**i**) binding of fibrinogen with PAA/Au nanoparticles (**a**) Binding of 7 nm nanoparticle to fibrinogen revealing each protein molecule accommodated two nanoparticles (**b**) 10–12 nm-sized nanoparticles prevent the binding of two particles to each fibrinogen due to the flexibility of fibrinogen at E domain of protein resulting the contact of second binding site with the nanoparticle (**c**,**d**) Larger nanoparticles (15–22 nm) can accommodate multiple fibrinogen molecules due to the larger surface area (**ii**) TEM of PAA-CS nanoparticles at (**a**) pH = 4.5 and (**b**) at pH = 7.4. (**iii**) Morphology of PAA-CS nanoparticles synthesized by the various processes at 4.5: (**a**) CS dropping into PAA solution; (**b**) PAA dropping into CS solution, (Reprinted from Refs. [81,83] with permission).

**Figure 5 polymers-14-01259-f005:**
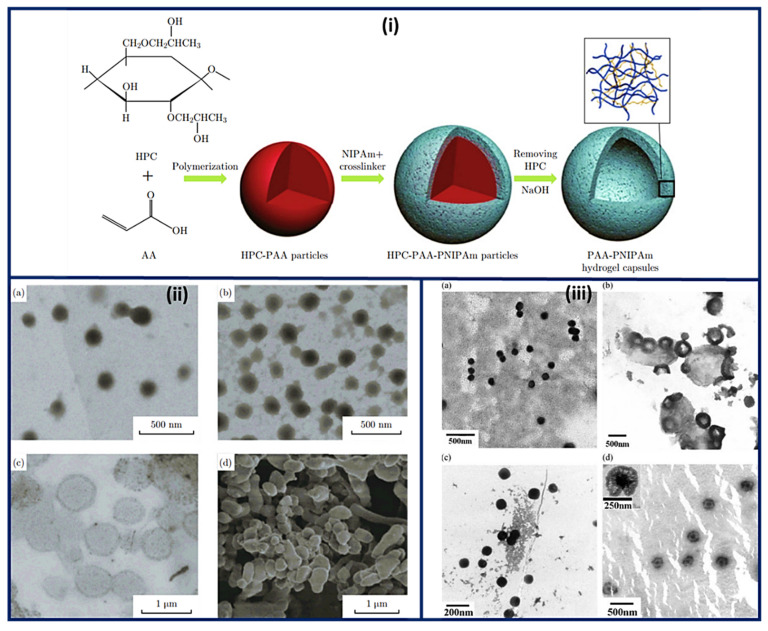
Schematic representation of the (**i**) synthesis and architecture of PNIPAm-PAA hydrogel capsules (**ii**) TEM micrographs of (**a**) PAA-HPC particles (pH 2.4) (**b**) PNIPAm-PAA-HPC composites (pH 2.4), (**c**) PNIPAm-PAA hydrogel capsules (pH 8.0) and (**d**) SEM image of PNIPAm-PAA hydrogel capsules after freeze-drying procedure. (**iii**) Morphology of nanosphere and nanocapsule of PAA/BSA (**a**) PAA/BSA, (**b**) nanocapsule of PAA/BSA, (**c**) PAA/BSA/GA, and (**d**) nanocapsule of PAA/BSA/GA. (Reprinted from Refs. [90,91] with permission).

**Figure 6 polymers-14-01259-f006:**
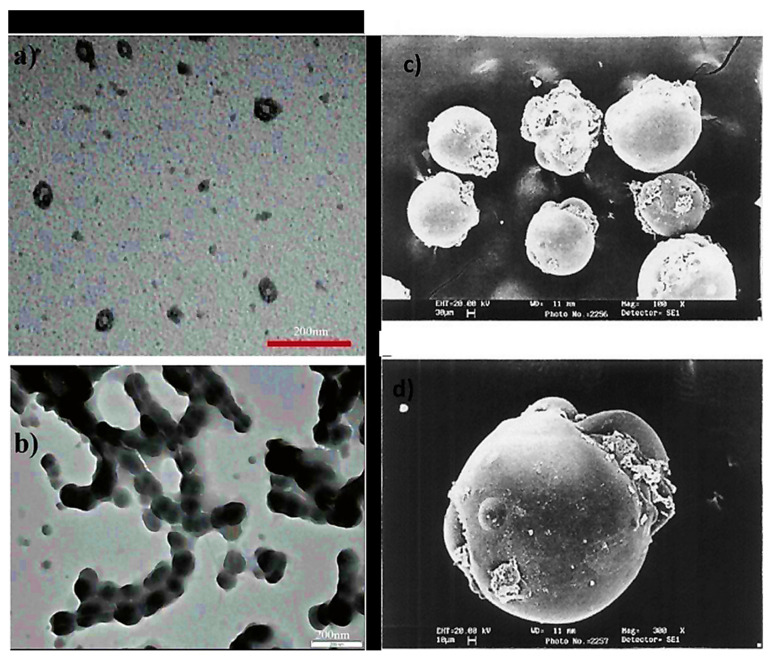
TEM micrographs of (**a**) cross-linked PAA-b-PI micelles covered with calcium phosphate, (**b**) PAA nanocages-Calcium phosphate displaying enhanced levels of aggregation and mineralization (after eight months) (**c**) SEM microspheres of PAA without forming agglomerations (spherical) (**d**) The smooth surface of microspheres without any pores. (Reprinted from Refs. [95,96] with permission).

**Figure 7 polymers-14-01259-f007:**
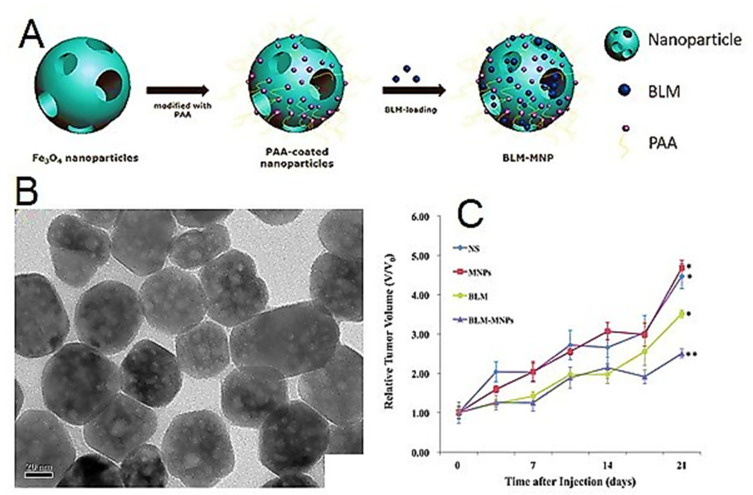
(**A**) Step-by-step synthesis of magnetic nanoparticles with PAA coating in the outer layer and BLM molecules adorned with PAA is depicted in this figure. (**B**) TEM image of bare nanoparticles. (**C**) The therapeutic effect of BLM-MNPs on tumors under the magnetic field. After several treatments, the relative tumor volume of several groups of mice showed (*n* = 6): mice injected with NS, MNPs, BLM, and BLM-MNPs [104].

**Figure 8 polymers-14-01259-f008:**
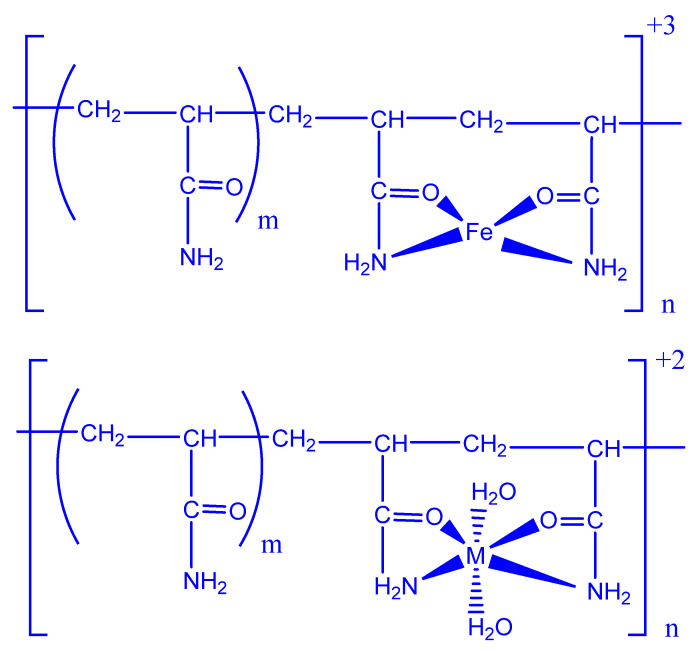
The proposed interaction between PAA and metal.

**Figure 9 polymers-14-01259-f009:**
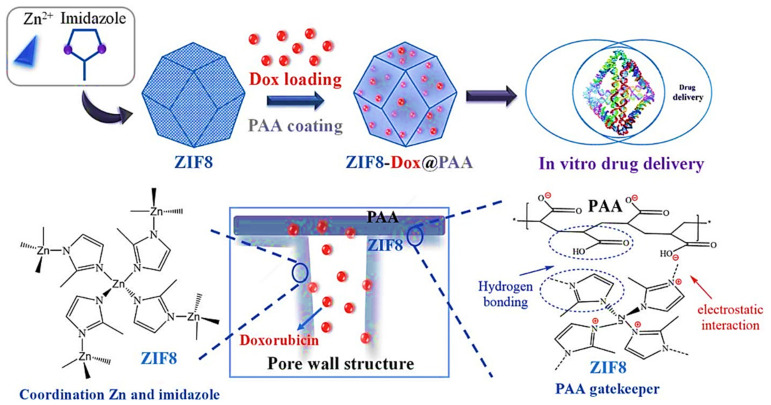
The preparation of ZIF8–Dox@PAA is depicted schematically [122].

**Table 1 polymers-14-01259-t001:** PAA-based core-shell nanostructures on a variety of substrates.

Substrates	Substrate Form	Refs.
Fe_3_O_4_	nanoparticles	[131,132,133,134,135,136,137,138,139]
Fe_3_O_4_	nanogels	[140]
Fe_3_O_4_	ferrogels	[141]
Fe_3_O_4_	hydrogel	[142]
γ-Fe_2_O_3_	nanoparticles	[143]
CoFe_2_O_4_	nanoparticles	[144,145]
NiFe_2_O_4_	nanoparticles	[146]
Multi-walled carbon nanotubes	nanocomposite	[122]
Mn_3_O_4_	nanoparticles	[147]
MnCO_3_	microcapsules	[148]
Co_9_S_8_@MnO_2_	nanoparticles	[149]
Fe_3_O_4_@MnO_2_	nanoparticles	[150]
Fe_3_O_4_@MnO_2_-doped NaYF4:Yb/Er/Nd	nanosheets	[151]
Au NPs	nanoparticles	[152,153,154,155]
Au NPs	hydrogel	[156]
Au NPs	nanoclusters	[157]
Au NPs	nanorods	[158]
SiO_2_	nanoparticles	[159,160,161,162,163]
Au NPs@SiO_2_	rubber film	[164]
MOF	nanoparticles	[165]
rGO	hydrogel	[166]
Mg-Ca_3_(PO_4_)_2_	clusters	[167]
TiO_2_	nanoparticles	[168,169]
CaCO_3_	nanoparticles	[170]
CeO_2_	nanoparticles	[171]

## Data Availability

The data presented in this study are available on request from the corresponding author.

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
