# Peer review of "Polyacrylic Acid Nanoplatforms: Antimicrobial, Tissue Engineering, and Cancer Theranostic Applications"

_polymers, 2022, doi:10.3390/polym14061259_

Round 1
Reviewer 1 Report
Ms. Ref. No. polymers-1615295-peer-review-v1
Title: Polyacrylic Acid Nanoplatforms: Synthesis, Architectures and Biomedical Applications
Polymers
The authors conducted research work on the topic of “Polyacrylic Acid Nanoplatforms: Synthesis, Architectures and Biomedical Applications”. The results are proper for publishing in this journal after some justification and some issues must be addressed before its recommendation for publication, such as:
- An abstract is normally a comprehensive summary of the published literature and adds the importance of the described methods. Revise the whole abstract part addressing problems and their solution using PAA for biomedical applications.
- Update the abstract with result outcomes of the best formulations that increase the interest of reviewers.
- Try to add one more paragraph about PAA for biomedical applications
- Try the following published manuscripts to describe more information on PAA-based polymeric composite materials for biomedical applications.
- https://doi.org/10.1016/j.arabjc.2020.102924
- Please do consider the following well-described literature in 1. Metal Oxide section to give interaction between the metal oxide and PAA polymeric matrix.
-
- https://doi.org/10.3390/polym12061238
- Please replace old references with current investigations and outdated references, especially the last five years.
- It is better to separate the future perspective and conclusion section
- Extensive editing of the English language (grammar, sentence making, spelling, typo errors, etc.) is required.
- The format and font style of the manuscript is not uniform and try to make it uniform also update Polymers reference style.
- Remove the list of abbreviations and try to define all the abbreviations, while using the first time.
Author Response
Reviewer #1 (Yellow Highlighted)
The authors conducted research work on the topic of “Polyacrylic Acid Nanoplatforms: Synthesis, Architectures and Biomedical Applications”. The results are proper for publishing in this journal after some justification and some issues must be addressed before its recommendation for publication, such as:
1- An abstract is normally a comprehensive summary of the published literature and adds the importance of the described methods. Revise the whole abstract part addressing problems and their solution using PAA for biomedical applications. Update the abstract with result outcomes of the best formulations that increase the interest of reviewers.
Answer: Thanks for your suggestion, the whole abstract was revised based on your advice.
2- Try to add one more paragraph about PAA for biomedical applications.
Answer: We have included one more paragraph about PAA in the biomedical application section.
3- Try the following published manuscripts to describe more information on PAA-based polymeric composite materials for biomedical applications.
https://doi.org/10.1016/j.arabjc.2020.102924
Answer: PAA-based polymeric composites are included in the tissue engineering section.
4- Please do consider the following well-described literature in 1. Metal Oxide section to give interaction between the metal oxide and PAA polymeric matrix.
https://doi.org/10.3390/polym12061238
Answer: We have now included the proposed scheme for PAA-metal interactions.
5- Please replace old references with current investigations and outdated references, especially the last five years.
Answer: outdated references were replaced with updated ones.
6- It is better to separate the future perspective and conclusion section
Answer: Future perspective was separated from conclusion section.
7- Extensive editing of the English language (grammar, sentence making, spelling, typo errors, etc.) is required.
Answer: All manuscript was double-checked to correct any language and structure errors.
8- The format and font style of the manuscript is not uniform and try to make it uniform also update Polymers reference style.
Answer: Thanks for your comment. The reference style and format of the paper were revised based on the Polymer template.
9- Remove the list of abbreviations and try to define all the abbreviations, while using the first time.
Answer: The abbreviation List was removed and incorporated in the text.
Reviewer 2 Report
The manuscript described synthesis, architectures, and biomedical applications of poly (acrylic acid) (PAA). The authors introduced the merits and characters of PAA. Thus, these findings will be useful for carriers of drug delivery. Therefore, the manuscript is not too excellent to be published. In other words, the manuscript is so excellent that it should be published.
Comments
(1) What sizes of nanoparticles are suitable for drug delivery as carriers?
(2) PAA is negatively charged. How does it interact with negatively charged heparan sulphates on proteoglycans or mucin?
(3) In line of 353, “Nanopcapsule” may be “Nanocapsule”.
That is all
Author Response
Reviewer #2 (Green Highlighted)
The manuscript described synthesis, architectures, and biomedical applications of poly (acrylic acid) (PAA). The authors introduced the merits and characters of PAA. Thus, these findings will be useful for carriers of drug delivery. Therefore, the manuscript is not too excellent to be published. In other words, the manuscript is so excellent that it should be published.
1- What sizes of nanoparticles are suitable for drug delivery as carriers?
Answer: In this review, our purpose is not to discuss the effect of nanoparticle size on drug delivery, but to answer the question of the respected referee, discuss It here “the size effect of nanoparticles The size of the NPs has an impact on their removal from circulation. Particles with size less than 5-6 nm are quickly cleared by the kidneys, while bigger particles with diameters greater than 200 nm are quickly cleared by the liver and spleen. The mononuclear phagocytic system removes particles that are 200 nm or larger (liver, spleen, and bone marrow)[i].[ii]. Additionally, these NPs should not cross the vessel walls in normal tissues thereby causing adverse effects. As the pore size of normal vessels is between 6 and 12 nm, this would suggest that nanoparticles should be larger than that size [iii]. The fundamental design criterion is that NPs must be able to pass through the pores of leaky tumor vasculature but not normal vascular pores. The pores of tumor arteries are typically between 40 and 200 nanometers in diameter[iv]”.
2- PAA is negatively charged. How does it interact with negatively charged heparan sulphates on proteoglycans or mucin?
Answer: Thanks for your valuable question. We bilevel, when it comes to proteoglycans or mucin interactions, PAA can be modified with positive charge inducer such as CTAB, STA, …. To have more interaction with negatively charged heparan sulfates on proteoglycans or mucin.
3- In line of 353, “Nanopcapsule” may be “Nanocapsule”.
Answer: Thank you, the comment was considered.
That is all
[i] . M. Moghimi, A. C. Hunter, and J. C. Murray, “Longcirculating and target-specific nanoparticles: theory to practice,” Pharmacological Reviews, 53, 2, 283– 318, 2001
[ii] M. Longmire, P. L. Choyke, and H. Kobayashi, “Clearance properties of nano-sized particles and molecules as imaging agents: considerations and caveats,” Nanomedicine, 3, 5. 703–717, 2008
[iii] H. Sarin, “Physiologic upper limits of pore size of different blood capillary types and another perspective on the dual pore theory of microvascular permeability,” Journal of Angiogenesis Research. 2, 1, 14, 201
[iv] Physical Properties of Nanoparticles That Result in Improved Cancer Targeting
Round 2
Reviewer 1 Report
The manuscript can be accepted after editorial check and some minor English correction that can be addressed during proof reading.